# Analysis of cfDNA fragmentomics metrics and commercial targeted sequencing panels

Kyle T. Helzer[1,12], Marina N. Sharifi [2,3,12], Jamie M. Sperger[3], Matthew R. Chrostek[1], Matthew L. Bootsma[1], Shannon R. Reese[1,3], Amy Taylor[3], Katie R. Kaufmann[3], Hannah Krause [3], Jennifer Schehr[2], Nan Sethakorn[2,3], David Kosoff[2,3], Christos E. Kyriakopoulos [2,3], Michael Bassetti[1,2], Grace Blitzer[1,2], John Floberg[1,2], Martin Sjöström [4], Andrew J. Armstrong [5], Himisha Beltran [6], Rana R. McKay [7], Felix Y. Feng [8], Ruth O'Regan[2,3,9], Kari B. Wisinski[2,3], Hamid Emamekhoo[2,3], Alex W. Wyatt [10,11], Joshua M. Lang [2,3,13] & Shuang G. Zhao [1,2,13] ✉

Fragmentomics based analysis of cell-free DNA (cfDNA) has recently emerged as a method to infer epigenetic and transcriptional data. Many of these reports analyze whole genome sequencing (WGS) which is not readily available clinically. Targeted exon panels are used for clinical cfDNA variant calling. In this report, we conduct an investigation of multiple published fragmentomics methods for WGS, but on cancer exon panels. We find that strategies utilizing normalized depth metrics, as well as all exons present on the panel, generally allow for better prediction of cancer phenotypes across a range of tumor fractions, though other metrics work particularly well in specific applications. Additionally, genes from commercial clinical targeted sequencing panels could be similarly employed for cancer phenotyping with a minimal decrease in performance despite their smaller genomic coverage. These results suggest that fragmentomics-based analysis of cfDNA can utilize targeted sequencing panels and does not necessarily require additional WGS.

Cell-free DNA (cfDNA) and its tumor-specific subset, circulating tumor DNA (ctDNA), are released from both benign and cancerous cells into the blood[1]. Variant calling from cfDNA is now clinically available, with multiple commercial targeted sequencing panels currently available for use[2–6]. Additionally, the proportion of cfDNA that is ctDNA can be calculated from these panels and carries important prognostic information[7–9]. An emerging approach in the cfDNA field is the study of DNA fragmentation patterns from tumor cells, also known as "fragmentomics". It has been found that the digestion and fragmentation of DNA in the process of cell death is not random, and that these patterns can be utilized to infer characteristics about the cell of origin, such as tumor type or subtype[10,11]. The size of DNA fragments is highly dependent on the placement of histones on DNA, which protect the DNA from degradation by DNAses in the blood[10,12]. The most frequent size of DNA observed in cfDNA is approximately 167 bp, which corresponds to the wrapping of DNA around a single histone complex[10,13,14].

[1]Department of Human Oncology, University of Wisconsin, Madison, WI, USA. [2]Carbone Cancer Center, University of Wisconsin, Madison, WI, USA. [3]Department of Medicine, University of Wisconsin, Madison, WI, USA. [4]Lund University, Lund, Sweden. [5]Duke Cancer Institute Center for Prostate and Urologic Cancers, Department of Medicine, Duke University, Durham, NC, USA. [6]Lank Center for Genitourinary Oncology, Dana-Farber Cancer Institute, Boston, MA, USA. [7]Moores Cancer Center, University of California San Diego, La Jolla, CA, USA. [8]Department of Radiation Oncology, Helen Diller Family Comprehensive Cancer Center, University of California San Francisco, San Francisco, USA. [9]Department of Medicine, University of Rochester, Rochester, NY, USA. [10]Department of Urologic Sciences, Vancouver Prostate Centre, University of British Columbia, Vancouver, BC, Canada. [11]Michael Smith Genome Sciences Centre, BC Cancer, Vancouver, BC, Canada. [12]These authors contributed equally: Kyle T. Helzer, Marina N. Sharifi. [13]These authors jointly supervised this work: Joshua M. Lang, Shuang G. Zhao. ✉e-mail: shgzhao@humonc.wisc.edu

Other complexes such as transcription factors and transcription machinery can also protect DNA from degradation[15–18], resulting in unique fragmentation patterns specific to genomic locations where these complexes are bound.

Much of the recent work on fragmentomics has focused on whole-genome sequencing (WGS), which provides a genome-wide snapshot of fragmentation patterns. While useful in certain contexts, WGS-based fragmentomics approaches are not applicable to the cancer exon panels used clinically, which require very high depth for variant identification. We recently reported on an approach for fragmentomics in standard cfDNA targeted exon sequencing panels[19]. Our analysis found that measuring the entropy of fragmentation sizes at the exon nearest to a gene's transcription start site allowed for the prediction of cancer types[19]. However, numerous other methods have been described analyzing fragmentation patterns using WGS, including nucleosome positioning[20], fragment size distributions[21], fragment counts[18,22], and DNA fragment end motifs[23], as well as fragment patterns around transcription factor binding sites[17] and open chromatin regions[24]. These metrics could also be applied to individual genomic regions like exons, but a comprehensive comparison of different WGS fragmentomics metrics in targeted exon panels has yet to be explored.

In this report, we examine exactly this across two separate cohorts for their ability to predict cancer samples vs. non-cancer samples, between various cancer types, and between subtypes in breast (ER-positive vs. ER-negative), prostate (adenocarcinoma vs. neuroendocrine), and lung cancers (small cell vs. non-small cell). Additionally, as commercial panels have separate and sometimes much smaller sets of genes assayed, we tested whether a subset of our data containing only genes present in each commercial panel would impact predictive performance. Finally, given that the detection and categorization of cancers via cfDNA is dependent on the ctDNA fraction, we performed in silico mixing studies on a deeply sequenced cohort with known ctDNA fractions to assess the performance of these metrics at very low ctDNA fractions. Overall, this study represents a significant advancement in our ability to utilize cfDNA fragmentomics to infer cancer phenotypes on targeted cancer exon panels such as the commercial panels already in use in the clinic.

## Results

### Overview of Fragmentomic Analyses

The standard use of cfDNA targeted cancer gene exon panels is for the detection of mutations and other DNA alterations originating from tumor cells. We recently published on the utility of fragmentomics from such targeted panels for cancer phenotyping, with a focus on the entropy in the first exons (E1) of genes in these panels[19]. Given the variety of methods for examining fragmentation patterns that have been reported, we sought to expand our analysis to both include regions beyond the first exon as well as to compare multiple classes of metrics which have been utilized for fragmentomic analysis of whole-genome sequencing. These include (A) fragment length proportions, which encompass fraction of small fragments (<150 bp)[21], the fraction of fragments in various bins of bp sizes, and diversity metrics like Shannon entropy which measure the spread of fragment sizes in a region[18,19], (B) normalized fragment read depth, which are the fragment counts normalized to both sequencing depth and the size of the regions analyzed[18], (C) end motif diversity score (MDS)[23], which quantifies the variation in 4-mer end motifs among fragments, (D) fragments overlapping with various transcription factor binding sites (TFBS)[16,17] and calculating their fragment size diversity, and (E) fragments overlapping with open chromatin sites[24] as defined by cancer-specific ATAC-seq data from TCGA and calculating their fragment size diversity (Fig. 1). These metrics were calculated at the exon level, with the exception of the TFBS and open chromatin analysis, which were performed on reads overlapping each transcription factor, or

cancer-specific open chromatic region, respectively. A total of 13 fragmentomics metrics were assessed: normalized depth at (1) all exons individually, (2) full genes, and (3) E1, Shannon entropy at (4) all exons, and (5) E1, MDS at (6) all exons, and (7) E1, small fragments at (8) all exons, and (9) E1, (10) fragment bins, (11) TFBS entropy, (12) ATAC entropy, as well as (13) all metrics combined. Each feature was then used to predict cancer type and subtype using a GLMnet elastic net model with 10-fold cross validation, repeated 25 times with different seeds.

### Fragmentomics Metric Performance in UW and GRAIL Cohorts

We performed our analysis in two independent cohorts – an in-house cohort collected at the University of Wisconsin (UW) comprising 431 samples (38 bladder cancer, 71 ER-positive breast cancer, 29 ER-negative breast cancer, 15 neuroendocrine prostate cancer (NEPC), 144 prostate adenocarcinoma, 44 renal cell carcinoma (RCC), 16 small cell lung cancer (SCLC), 47 non-small cell lung cancer (NSCLC), and 27 healthy i.e. non-cancer donors) sequenced on a custom 822-gene targeted exon panel covering roughly 2.4 Mb[19] at an average sequencing depth of 3038x, and a publicly available dataset from Razavi et al. comprising 198 samples (48 breast, 49 lung, 54 prostate, 47 healthy i.e. non-cancer; cancer subtype information not available) sequenced on a 508-gene targeted panel from GRAIL covering approximately 2.0 Mb at a sequencing depth >60000×[25] (Fig. S1). Differences in cohort details have been outlined previously[19]. 111 samples were added to the previously published UW cohort. By utilizing two different panels with different designs and depths, we hoped to identify panel-independent generalizable conclusions.

In the UW cohort, we found that normalized fragment read depth across all exons provided the best average performance for predicting cancer types and subtypes with an average AUROC of 0.943 and a range of 0.873 for NSCLC to 0.986 for healthy samples (Fig. 2; Table S1). We also observed that the other depth-based metrics performed nearly as well. First exon (E1) depth yielded an average AUROC of 0.930 and a range of 0.838 for NSCLC to 0.989 for healthy samples, while full gene depth (combining all exons from one gene into a single feature) yielded an average AUROC of 0.919 with a range of 0.828 for NSCLC to 0.993 for NEPC (Table S1). In the GRAIL cohort, we similarly found that normalized fragment read depth across all exons was the best performing metric with an average AUROC of 0.964 and a range from 0.914 for lung cancer to 1.000 for healthy samples (Fig. 3; Table S2). Interestingly, the top-performing metric for SCLC in the UW cohort was MDS across all exons with an average AUROC of 0.888, indicating that some features may be more suitable for certain cancer types. In general, for features where both the first exons and all exons were analyzed, the full set of exons generally performed as well or better compared to just the first exon alone in both cohorts, suggesting that fragmentomics patterns in these downstream exons may contain additional information (Table S1, S2). The fragmentomics metrics overall were not strongly correlated with GC content (Fig. S2). We also evaluated the performance of the Griffin algorithm[17] in comparison to the other fragmentomics metrics, and while it had superior performance compared to TFBS entropy in certain situations, it was worse in others (Fig. S3). Overall, these results demonstrate that selecting the optimal fragmentomics metrics can have a large effect on ultimate performance when using cancer gene-targeted panels, with a normalized read depth at each individual exon providing the strongest predictive power overall.

### Fragmentomics Metric Performance with Commercial Panel Genes

While fragmentomics demonstrates strong predictive value in these cohorts, the targeted panels used have a wider range of genes compared to several clinically utilized commercial panels. This greater

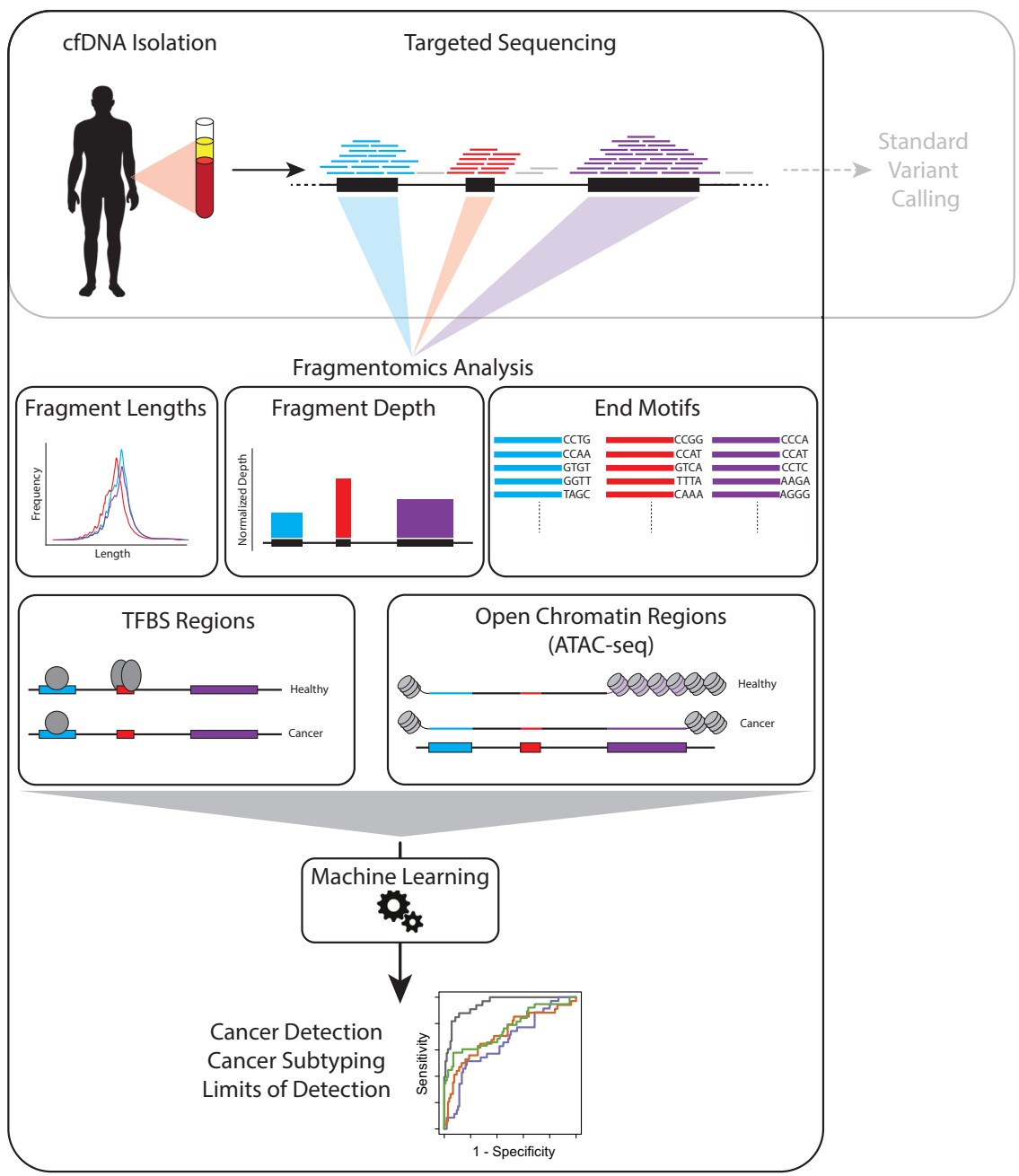

**Fig. 1 | Overview of Fragmentomics Methods. cfDNA is isolated from cancer patients and sequencing using a targeted exon sequencing panel.** Typically, this data is used for variant calling, but the reads overlapping each exon in the panel contain fragmentation information relevant to the tumor of origin. Fragmentation patterns of these reads were analyzed in five broad categories of Fragmentomic Analysis Classes (1) fragment lengths, (2) fragment depth, (3) fragment end motifs, (4) fragments overlapping with TFBS, and (5) fragments overlapping with open chromatin regions unique to various cancer types. These features were then used to train a machine learning model for cancer detection and subtyping. Human outline obtained from https://commons.wikimedia.org/wiki/File:SVG_Human_Silhouette.svg.

range of genes (UW panel 822 genes; GRAIL panel 508 genes) could be contributing to the strong performance of the fragmentomics models. We therefore wanted to assess if performance differed when restricted to the smaller gene sets present on commercial panels. To test if the genes present in commercially used panels could also serve as a predictive model, we extracted the subset of genes present in three commercial cfDNA panels (Tempus xF, 105 genes; Guardant360 CDx, 55 genes; and FoundationOne Liquid CDx, 309 genes; two genes from FoundationOne Liquid CDx with non-exonic coverage, *TERT* and *TERC*, were not included) from our data, and performed the same analysis as above. Greater than 90% of genes on each commercial panel were available in our datasets, with the exception of FoundationOne Liquid CDx in the UW panel, where only 77% of genes were available (Fig. S4).

In the UW cohort, we found that the FoundationOne Liquid CDx panel yielded the best overall performance of the three commercial panels tested, followed by Tempus xF and Guardant360 CDx (Fig. 4A, Fig. S5). Certain cancer types maintained high AUROCs across many features, with prostate adenocarcinoma retaining AUROCs >0.9 in most cases (Table S1). Interestingly, we found that depth at all exons still retained much of its predictive performance in the UW cohort across all three commercial panels, with a change in median AUROCs of −0.002 for FoundationOne Liquid CDx, −0.005 for Guardant360 CDx, and −0.012 for Tempus xF across all phenotypes (Fig. 4B, Fig. S5).

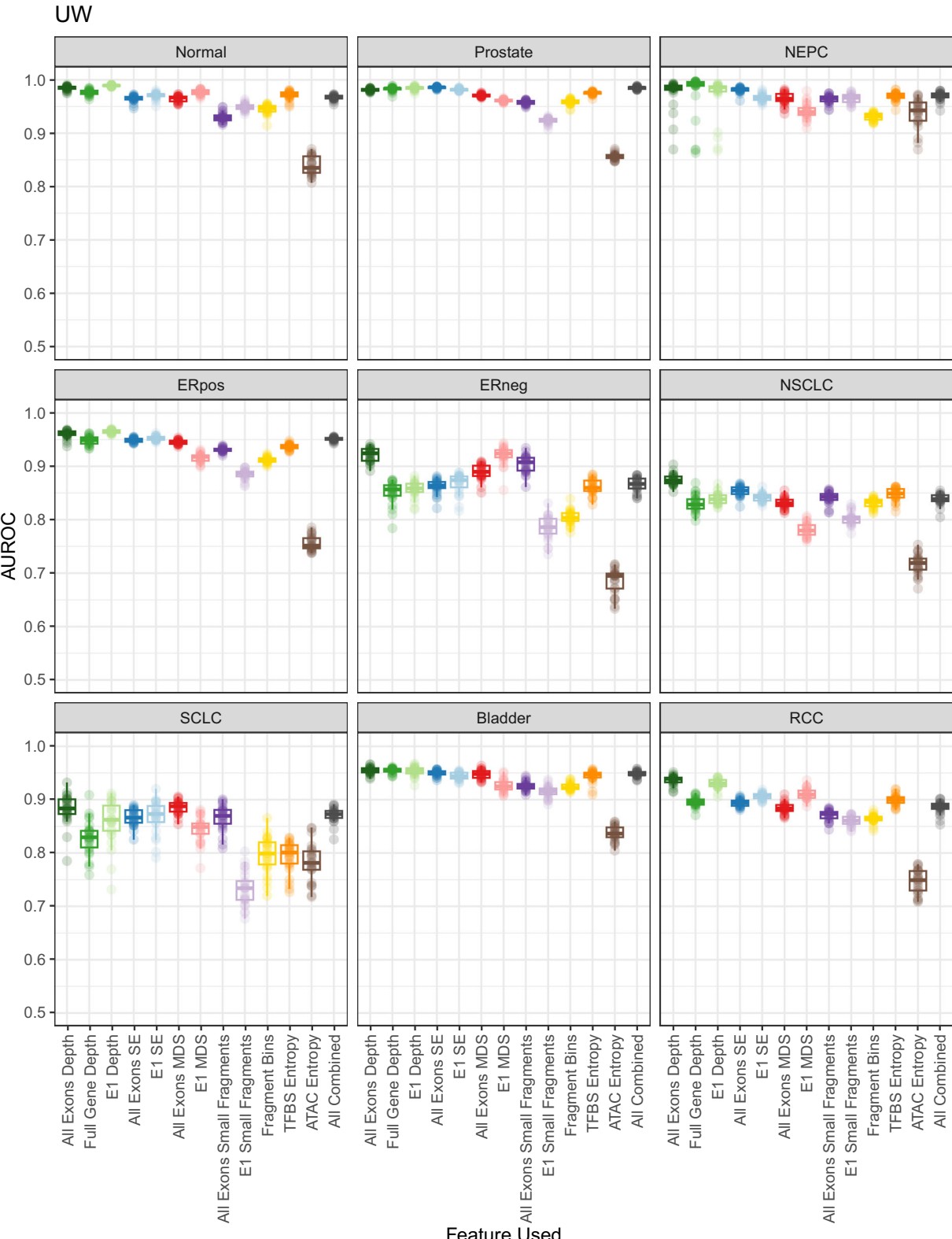

**Fig. 2 | Overview of Fragmentomic Metric Performance in the UW Cohort.** Various fragmentomics metrics from WGS studies were tested using an institutional targeted exon panel for their ability to predict eight different cancer types and subtypes along with healthy vs. cancer using a machine learning model with 10-fold cross validation. Twenty-five replicates were performed and boxplots of the AUROC for each metric are shown. NEPC neuroendocrine prostate cancer, ERpos ER-positive breast cancer, ERneg ER-negative breast cancer, NSCLC non-small cell lung carcinoma, SCLC small cell lung carcinoma, RCC renal cell carcinoma. Boxplots display the center as the median, with the bounds of the box as Q1 (25th percentile) and Q3 (75th percentile). Whiskers are defined by the lowest and highest value within 1.5 times the interquartile range (IQR; Q3 - Q1). Points outside of 1.5xIQR are displayed as individual points outside of the boxplot.

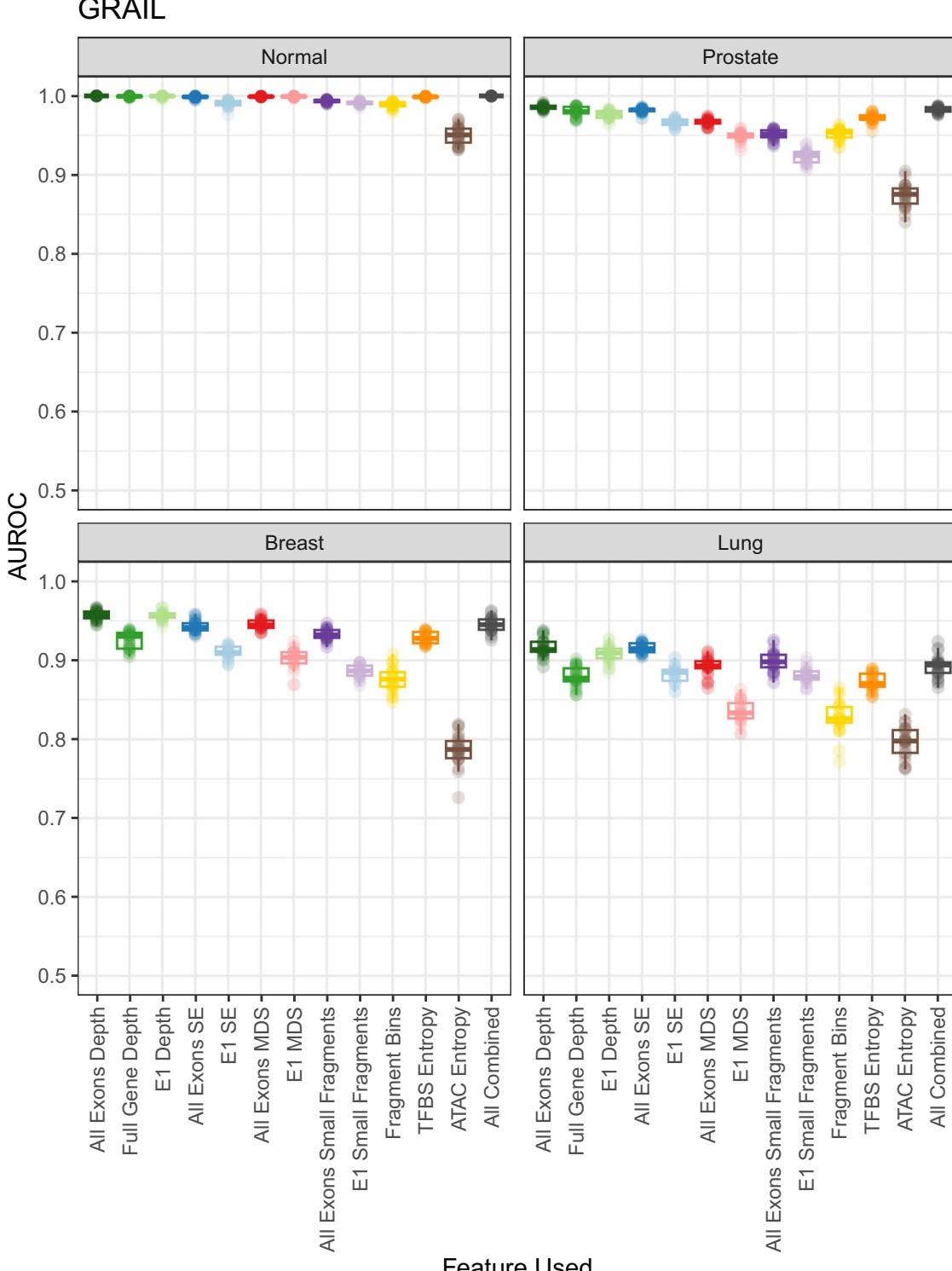

**Fig. 3 | Overview of Fragmentomic Metric Performance in the GRAIL Cohort.**
Various fragmentomics metrics from WGS studies were tested using a published targeted exon panel for their ability to predict three different cancer types along with healthy vs. cancer using a machine learning model with 10-fold cross validation. Twenty-five replicates were performed and boxplots of the AUROC for each metric are shown. Subtype information was not available for this cohort. Boxplots display the center as the median, with the bounds of the box as Q1 (25th percentile) and Q3 (75th percentile). Whiskers are defined by the lowest and highest value within 1.5 times the interquartile range (IQR; Q3 - Q1). Points outside of 1.5xIQR are displayed as individual points outside of the boxplot.

In the GRAIL cohort, all commercial panels and features tested showed strong performance in predicting healthy vs. cancer, with a range of AUROCs from 0.951 to 1.000 across all features (Fig. 5A and Fig. S5). Similar to the UW cohort, the GRAIL cohort models trained using the subset of genes in the FoundationOne Liquid CDx panel had the highest AUROCs compared to the Tempus xF and Guardant360 CDx genes. Likewise, the depth metric in all exons lost very little performance in the commercial panel gene subsets, with changes in median

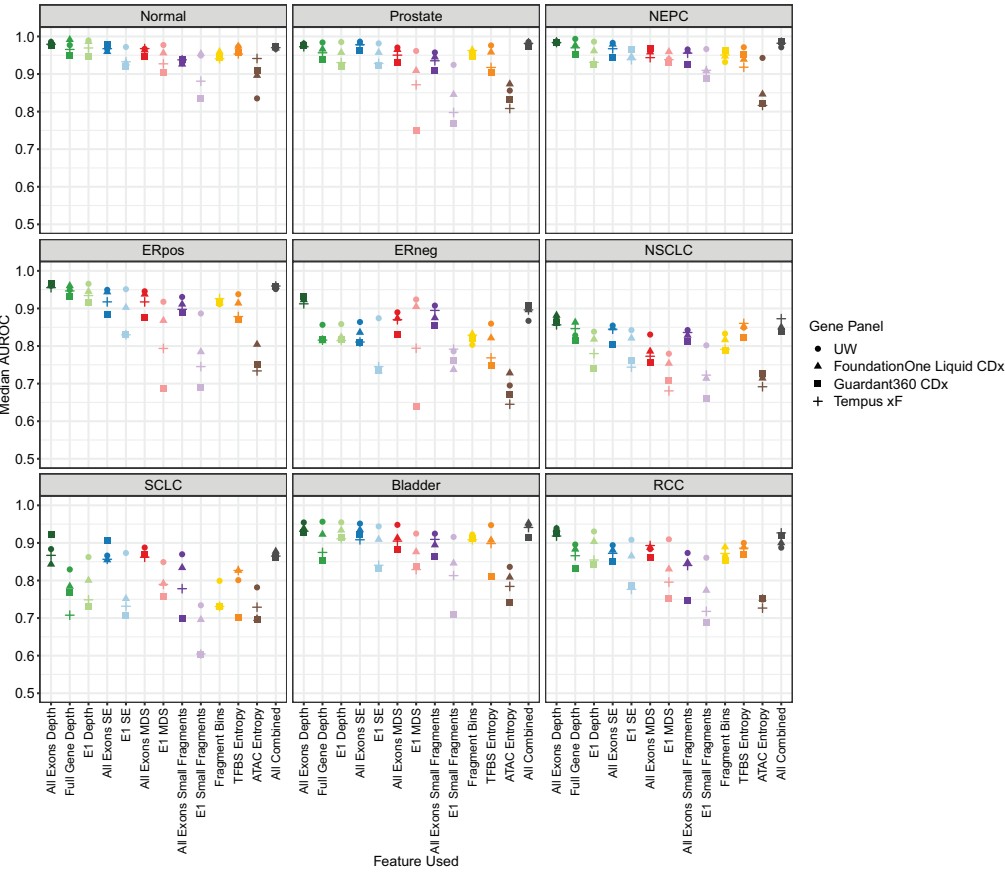

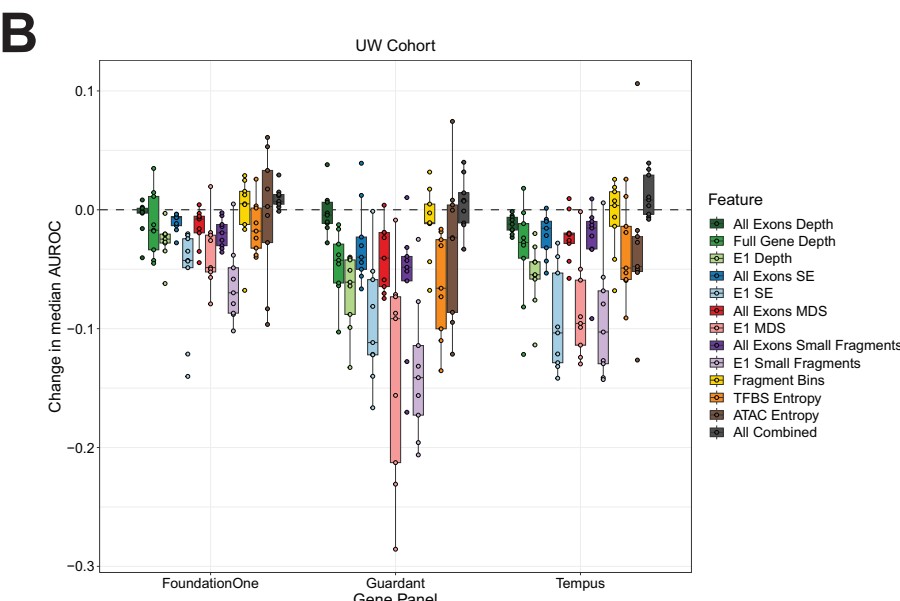

**Fig. 4 | Fragmentomics metric performance in commercial targeted panels in the UW cohort. A** Genes in common between the UW targeted panel and each respective commercial panel were used for feature selection. For each feature tested and for each gene panel, a 10-fold cross validation machine learning model trained to predict cancer type, subtype or healthy vs cancer. Each model was repeated 25 times and the average AUROC is reported. **B** Difference in median AUROCs between using all the genes in the UW targeted panel and the genes in

each respective commercial panel for model training in each feature. Boxplots represent the distribution of AUROC differences across the nine phenotypes tested and display the center as the median, with the bounds of the box as Q1 (25th percentile) and Q3 (75th percentile). Whiskers are defined by the lowest and highest value within 1.5 times the interquartile range (IQR; Q3 - Q1). Points outside of 1.5xIQR are displayed as individual points outside of the boxplot.

## A

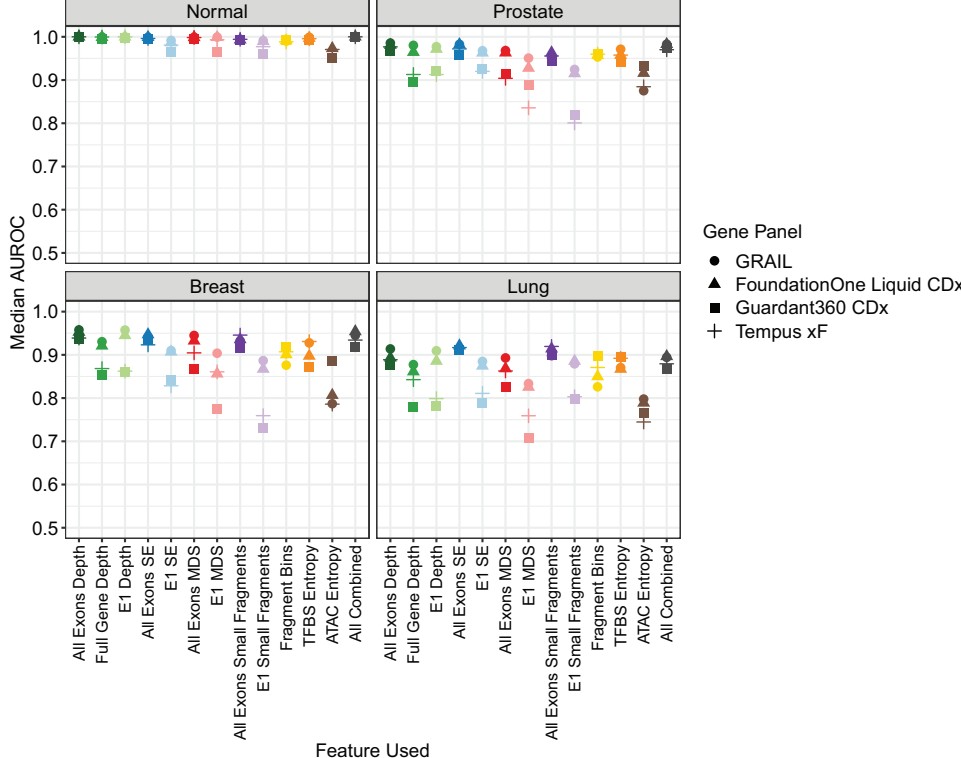

## B

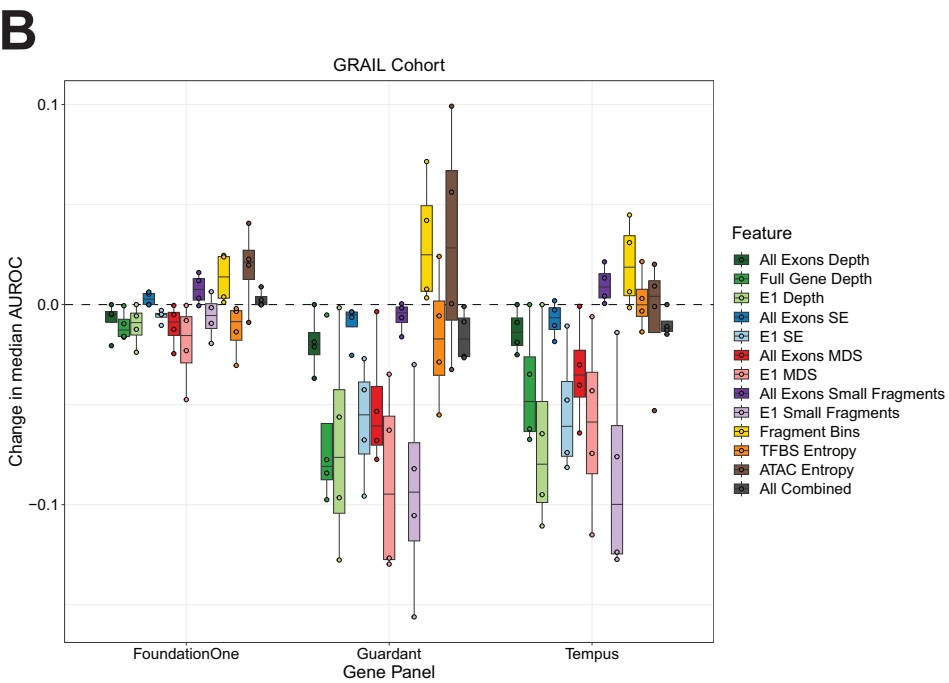

**Fig. 5 | Fragmentomics metric performance in commercial targeted panels in the GRAIL cohort. A** Genes in common between the GRAIL targeted panel and each respective commercial panel were used for feature selection. For each feature tested and for each gene panel, a 10-fold cross validation machine learning model was trained to predict cancer type or healthy vs cancer. Each model was repeated 25 times and the average AUROC is reported. **B** Difference in median AUROC between using all the genes in the GRAIL targeted panel and the genes in each respective commercial panel for each feature. Boxplots represent the distribution of AUROC differences across the four phenotypes tested and display the center as the median, with the bounds of the box as Q1 (25th percentile) and Q3 (75th percentile). Whiskers are defined by the lowest and highest value within 1.5 times the inter-quartile range (IQR; Q3 - Q1). Points outside of 1.5xIQR are displayed as individual points outside of the boxplot.

AUROCs of −0.005 for FoundationOne Liquid CDx, −0.020 for Guardant360 CDx, and −0.014 for Tempus xF across all phenotypes (Fig. 5B and Fig. S6). We also observed that many of the exon 1 metrics displayed a larger decrease in performance when compared to their all-exon counterparts (e.g. E1 MDS, E1 Small Fragments), particularly in the commercial panels that contain fewer genes like Guardant360 CDx (55 genes) and Tempus xF (105 genes) (Figs. 4B, 5B and Fig. S6). These data demonstrate that cancer vs. non-cancer and type/subtype prediction from commercially available targeted exon panels is feasible despite fewer genes, and could be applied to existing data already present in clinical cfDNA testing. However, selecting the appropriate fragmentomics metric is crucial.

**Targeted Cancer Panel Fragmentomics at Low ctDNA Fractions**
Many cfDNA samples have low ctDNA fractions, especially in earlier cancer stages or responding tumors. It is therefore important to understand the performance of fragmentomics approaches as a function of ctDNA fraction. To test how each feature performed at very low ctDNA fractions, we took advantage of the deep sequencing performed in the GRAIL cohort ( > 60000X raw sequencing depth) with tissue-informed variant calling for highly accurate ctDNA fraction estimates. We then performed computational mixing of cancer sample reads with healthy sample reads to create in silico very low ctDNA fraction samples at a uniform sequencing depth (Fig. 6A). Only cancer samples with available ctDNA fraction data (n = 105) and all healthy samples (n = 47) were used for this analysis. First, the cohort was split into a training and validation cohort before mixing with 70% of the cohort split into training and 30% split into validation. In the training cohort, each combination of healthy sample (n = 32) and cancer sample (n = 71) was mixed in silico at five separate ratios (50:50, 75:25, 90:10, 95:5, 99:1; 100 M reads total) by randomly selecting reads from each sample. As a control, each combination of healthy samples was mixed with each other in the same ratios to be used as synthetic healthy samples which helps avoid imbalanced classes during training. These 11360 mixed in silico "synthetic" cancer samples (32 ×71 x 5) along with the 4960 mixed healthy control samples (32 ×31 x 5) yielded a total of 16320 samples for training. Additionally, the remaining samples in the validation cohort were combined and mixed in the same manner, resulting in 2550 cancer (15 ×34 x 5) and 1050 (15 ×14 x 5) control samples for a total of 3600 validation samples, ensuring no data leakage between the training and validation cohorts. Then for each synthetic sample, we calculated the features used in this report and trained a model to predict cancer type. Given our conclusions above, that features at all exons were usually as good or better than the first exon alone, we restricted testing to the metrics calculated at all exons. This whole process was repeated 25 times with different seeds. Results were then split into bins based on their calculated ctDNA fraction after down-sampling and mixing, using the provided ctDNA fraction as the baseline[25], and then the AUROCs for each phenotype was calculated within each bin.

As expected, all features lost predictive power as the calculated ctDNA fraction decreased (Fig. 6B). At the highest ctDNA fraction bin (0.1 – 1), performance for differentiating healthy vs. cancer samples was near perfect for most metrics, which was anticipated given that these samples have the highest fraction of cancer reads. Depth at all exons yielded strong performance for breast and prostate cancer prediction, which mirrored our results from the analysis of unmixed samples. However, lung cancer prediction showed more variability likely due to the low number of unique lung cancer samples that fell into this ctDNA fraction bin (0.1 – 1) in this cohort. With decreasing ctDNA fraction, model performance for individual cancer types decreased more compared to predicting healthy vs. cancer[19]. At the lowest ctDNA fraction bin (0.0001 – 0.001), the AUROCs approached 0.500 for many features, as the healthy reads become the dominant source of information in the sample. Interestingly, even at this low

ctDNA fraction, depth at all exons, full gene depth, SE at all exons, and TFBS entropy maintained median AUROCs >0.70 for differentiating healthy vs. cancer, indicating that these fragmentomics metrics may still be identifying the tumor signal. The depth-based metrics performed well at this level for predicting healthy vs. cancer considering the low ctDNA fraction, with a median AUROC of 0.774 for depth at all exons. The performance also decreased as a function of sequencing depth when we performed computational down-sampling of the GRAIL cohort, though less so for some metrics (Fig. S7). These results emphasize the use of depth-based metrics, especially those using all exons, for fragmentomics analysis of targeted panels and demonstrate that cancer detection may still be possible at lower ctDNA fractions and sequencing depths.

## Discussion
The analysis of WGS cfDNA fragmentomics has resulted in a broad set of clinically and biologically relevant information that can be inferred, including histone occupancy[10,20], gene expression[18,22,26], and transcription factor binding[16,17,27]. We performed an evaluation of these WGS fragmentomics metrics in standard targeted cancer gene exon panels, demonstrating how a normalized depth metric using all the exons present in the sequencing panel generally worked best for predicting cancer type, subtype, and differentiating between healthy controls vs. cancer samples across panel designs. This result is likely due to multiple factors. Firstly, variations in relative depth can indicate the presence of somatic copy number alterations. We and others have also demonstrated that fragment coverage (i.e. depth) near the transcription start sites of highly expressed genes displays a decrease in cfDNA sequence coverage due to the lack of histone occupancy in these active regions, while the opposite is true for lowly expressed genes, and that this correlation extends into the early exons[18,19,22,27]. Similar patterns of fragment coverage have been described at exon-intron junctions as well[22], which could explain the better performance of depth at all exons separately compared to the full gene depth feature (where all of the exons are combined into one single feature). Measuring all these variables may provide the model with more granular information from which to accurately predict the cancer phenotype. Interestingly, for certain cancer types such as SCLC, non-depth-based metrics such as MDS seemed to perform remarkably well. In general, ATAC Entropy consistently underperforms the other metrics in both panels and all classes. This is likely due to less information being contained within ATAC peaks compared to all exons, especially since exons are not particularly enriched for regulatory regions. Combining all metrics into a single integrated model did not improve performance over the best individual metrics in the full panels, though there was perhaps a small improvement in performance in the smaller commercial panel gene subsets.

Depending on the cancer stage, type, and clinical status, ctDNA fractions can be quite low in cfDNA samples[28]. Therefore, it is critical to understand the performance of these metrics in this setting. However, this is difficult to benchmark in patient samples because the lower the ctDNA fraction, the more uncertainty there is in accurate estimations due to technical limitations of the assay and biological confounders such as clonal hematopoiesis[29]. We therefore used an in silico mixing approach paired with a high sequencing depth dataset with highly accurate tissue-guided ctDNA fraction estimates to accurately create "synthetic" low-ctDNA fraction samples. We found that the depth-based metrics, especially those using all exons, again provided the best results, even at low ctDNA fractions. Also, while the performance of all classifiers decreased with decreasing ctDNA fraction as expected, cancer vs. healthy non-cancer prediction accuracy was higher than cancer type predictions at low ctDNA fractions[19]. While these insights are important, the true clinical limit of detection and performance in low ctDNA fraction samples needs to be ideally confirmed using low

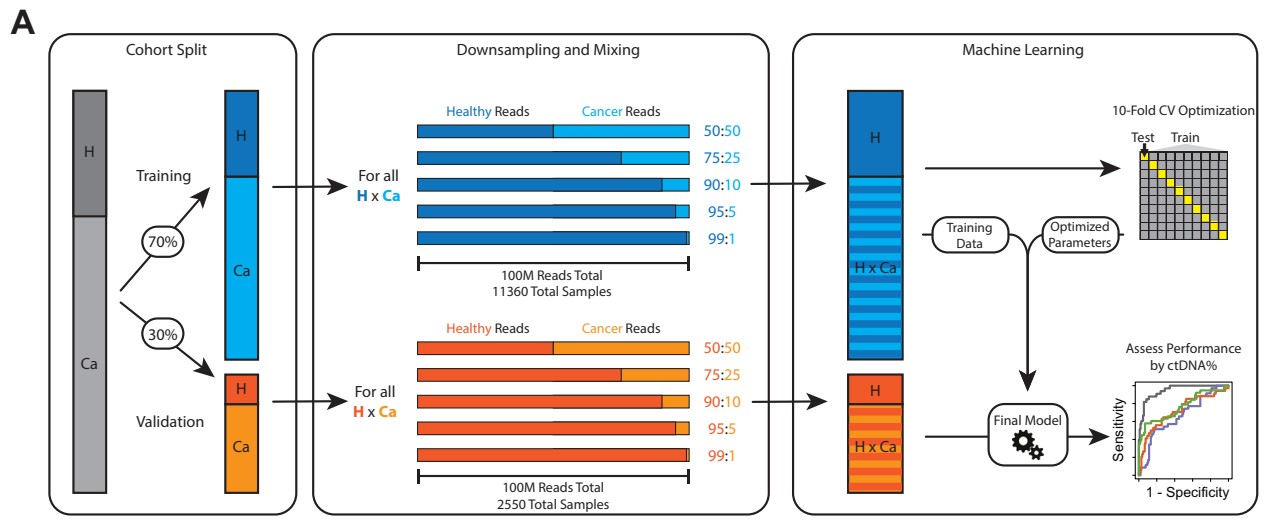

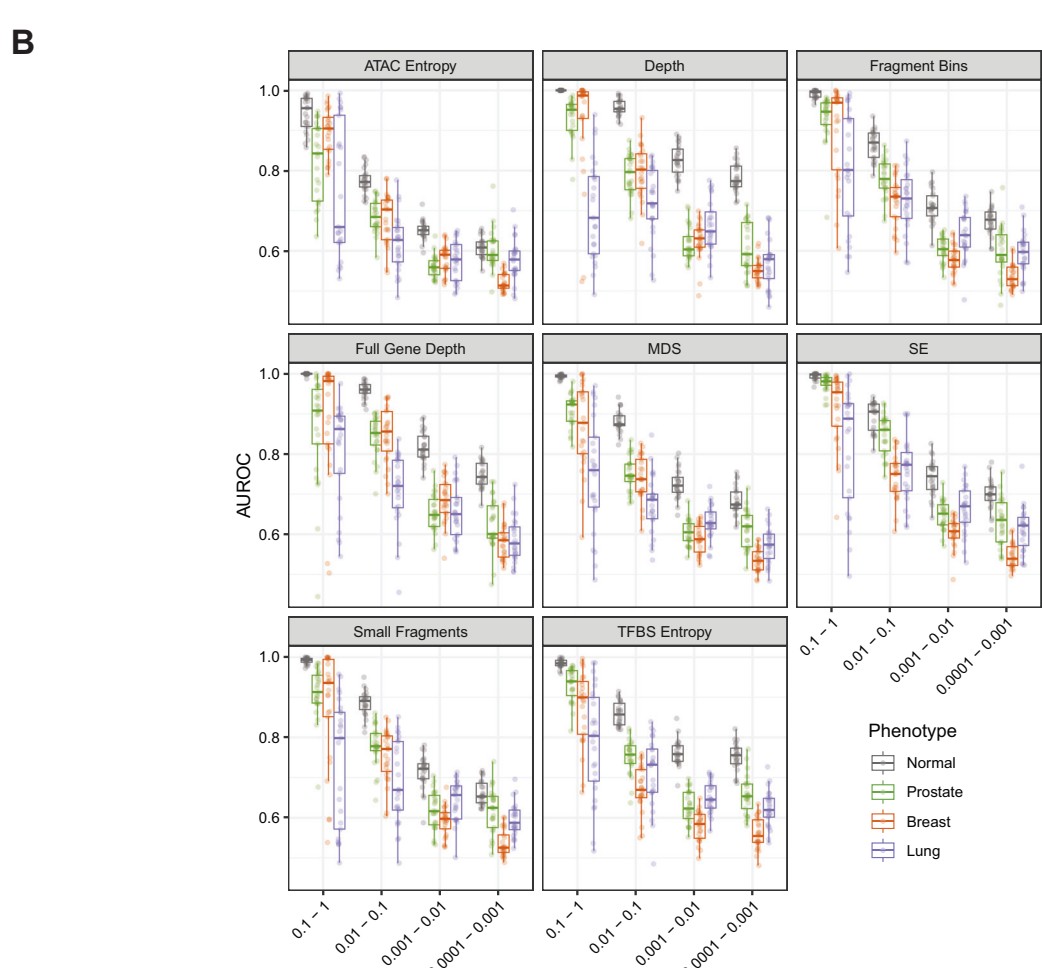

ctDNA fraction clinical samples, as in silico combinations do not perfectly replicate the full in vivo cfDNA environment.

Overall, this report represents a significant advancement in applying cfDNA fragmentomics to targeted exon sequencing panels, which unlike WGS, are already widespread in both research and clinical practice due to their ability to cost-effectively identify somatic mutations. Thus, the application of fragmentomics to these panels and

datasets can provide a great deal of additional utility with minimal additional cost or effort. Importantly, we demonstrate that our approach could be successfully utilized on commercial targeted cfDNA sequencing panels from Tempus, Guardant, and FoundationOne. Notably, this analysis was limited to genes from the commercial panels which overlapped with our tested panels, thus using the full set of genes in each commercial panel could potentially result in even

**Fig. 6 | In silico Mixing of Cancer and Healthy Samples for Low ctDNA Fraction Assessment. A** Samples in the GRAIL cohort were split into 70% training and 30% validation prior to down-sampling and mixing. For all healthy (H) and cancer (Ca) combinations in each respective cohort, reads were down-sampled and mixed at five different proportions (50:50, 75:25, 90:10, 95:5, and 99:1) for a total of 100 M reads. As a control, all healthy/healthy combinations were mixed in the same manner. The in silico-generated samples were then used to train a machine learning model to predict cancer type in the validation cohort using 10-fold cross-validation

with AUROC as the primary output for performance. This process was then repeated 25 times to yield a range of results for each feature. **B** Results of the machine learning model for each feature tested, binned by calculated ctDNA fraction based on the original ctDNA fraction of the cancer sample and the mixing ratio. Boxplots display the center as the median, with the bounds of the box as Q1 (25th percentile) and Q3 (75th percentile). Whiskers are defined by the lowest and highest value within 1.5 times the interquartile range (IQR; Q3 - Q1). Points outside of 1.5xIQR are displayed as individual points outside of the boxplot.

stronger performance. It is important to note that the goal of this study is not to produce and assess performance of a "locked" prediction model. Instead, we seek to compare different metrics to assess relative performance in a controlled and comprehensive manner. The absolute performance of an approach will depend on the exact clinical or biological question, panel design, sequencing depth, and other factors. However, relative performance can give a sense of the strengths and weaknesses of different fragmentomics metrics. Understanding the performance characteristics of these approaches is crucial as we transition this technology more fully onto clinical platforms. Most current targeted panels are based around cancer gene coding exons as they are primarily looking for DNA variants. Our work suggests that adding genes whose regulation/expression is of interest would allow for additional fragmentomic analysis.

## Methods

### Cohort Description

The University of Wisconsin (UW) cohort comprises metastatic cancer patients enrolled in a liquid biology collection protocol which has been approved by an institutional review board (2014-1214) in addition to samples from two ongoing clinical trials (NCT03090165, NCT03725761). Sample collection has complied with all relevant ethical regulations. Healthy blood samples were purchased from the University of Wisconsin Carbone Cancer Center (UWCCC) tissue BioBank and processed in the same manner as the cancer samples. Informed consent was obtained from all donors.

### Sequencing and processing of UW samples

Peripheral blood samples were collected and processed[19]. Briefly, cfDNA was extracted from 2-6 mL of plasma using the QIAamp Circulating Nucleic Acid Kit (Qiagen, Germantown, MD) which was isolated from an original 10 mL of peripheral blood. cfDNA was then subjected to library preparation using unique molecular identifiers (UMIs) followed by hybridization and capture with a custom 822-gene exon panel. The resulting libraries were subjected to paired-end sequencing (2 ×150 bp) on a NovaSeq 6000 platform (Illumina, San Diego, CA) at the UW Biotechnology Center at 50 M reads per sample. Reads were aligned to the hg38 genome with BWA-mem[30] (v0.7.17) and deduplicated using UMI tools[31] (v1.1.2) which utilizes unique molecular identifiers (UMIs) and start-stop positions. As samples were processed over several years, sequencing was done whenever a sufficient number of samples were available. 97% of samples were processed in batches that contained multiple phenotypes, though not all phenotypes were represented in every batch. All samples were processed in an identical manner using the exact same protocol. Further details of sequence data processing can be found in our previous report[19].

### Machine learning

For all machine learning performed in this report, a multinomial logistic regression using the generalized linear model with elastic net penalty (GLMnet) was used for cancer type and cancer vs. healthy predictions. In all models except the in silico mixing experiment, 10-fold cross-validation was used for training and validation, with 25 unique seeds resulting in 25 unique sets of 10 folds. Folds were set first so that the same folds could be used; therefore, training/validation was used for all fragmentomics

metrics, making direct comparison possible. For the in silico mixing experiment, 70% of the cohort was used for training, while the remaining 30% was withheld for independent validation. For all models, Latin hypercube sampling was utilized to select the range of α and λ parameters to be tested in training using the grid_latin_hypercube function from the R dials package (v.1.2.0) with 10 parameters, and performed nested cross-validation with 5 folds in the inner-fold. Additionally, up-sampling of imbalanced classes was performed to minimize biases. After training, the model with the highest area under the ROC curve (AUROC) was selected, and the parameters from this model were used for the final training model, which was then trained on the whole training cohort and used for validation. For all experiments, this procedure was repeated twenty-five times with different seeds to generate a range of results, which were then summarized. The medians across these replicates are reported.

### Metric calculations

Paired reads with fragment size inserts between 20 bp and 500 bp were retained for analysis. In all instances where reads are overlapped for analysis, bedtools32 intersect (v2.30.0) was used. A minimum overlap of 1 bp was required for a read to be considered as overlapping. If a read overlapped with multiple regions, it was used in the metric calculation for both regions. First exon feature tables were extracted after features were calculated at all exons. In all instances where reads are overlapped for analysis, bedtools[32] intersect (v2.30.0) was used. A minimum overlap of 1 bp was required for a read to be considered as overlapping. If a read overlapped with multiple regions, it was used in the metric calculation for both regions. First exon feature tables were extracted after features were calculated at all exons.

**Normalized read depth.** For each cfDNA sample, reads were overlapped with the exon regions for the respective panel used. For each exon, overlapping reads were totaled and normalized by dividing by the size of the exon in bp and then dividing by the total number of reads in the sample. For the first exon depth, the first coding exons were used from these features. If the first coding exon was not present on the panel for a gene, then the closest coding exon was substituted. Depth at all exons individually, the full gene, and only at E1 were investigated.

**Shannon entropy.** After overlapping of cfDNA reads with panel exons, fragment lengths were calculated, and the frequency and count of each fragment length was determined at each exon on the respective panels. Shannon entropy was calculated on the frequency of the fragment lengths using the 'entropy' package (v1.3.1) in R (v4.3.0). This yielded a single entropy value for each exon in each sample. Shannon entropy at all exons individually and only at E1 were investigated.

**Fragment bins and small fragments.** After overlapping of cfDNA reads with panel exons, fragments were binned based on the fragment length and the proportion of fragments in each bin was calculated. The fragment size bins were 0-100 bp, 101-150 bp, 151-200 bp, 201-250 bp, 251-300 bp, and greater than 300 bp. This yielded six features per exon in the respective panels for the fragment bin metric, which was further used in our investigations. For the small fragment metric, the

proportion of reads less than or equal to 150 bp was calculated, yielding one feature per exon. Small fragments at all exons, individually and only at E1, were investigated.

**Motif diversity score**. Motif diversity score was calculated as previously described[23]. Briefly, for the reads overlapping each exon, the 4-mer end motifs were extracted from both the 5' and 3' ends using bedtools getfasta and MDS was calculated on the distribution of the frequencies of the resulting 256 motifs. This yielded a single MDS score for each exon in the panel. MDS scores at all exons individually and only at E1 were investigated.

**TFBS entropy**. To assess the distribution of cfDNA fragments at transcript factor binding sites (TFBS), a collection of consensus *Homo sapiens* TFBSs was downloaded from the Gene Transcription Regulation Database[33] (GTRD, v19.10). Specifically, the 'Homo_sapiens_meta_clusters.interval' file was downloaded. For each TF, the top 5000 sites with the greatest amount of experimental support (exp count column) were used for analysis, which is defined as the number of experiments that detected the binding site. TFs with fewer than 5000 sites were discarded, leaving a total of 808 TFs used for the analysis. Read fragments from each cfDNA sample were then overlapped with the curated TFBSs, and Shannon entropy was calculated on the frequencies of the fragment sizes overlapping sets of TFBSs. This yielded an entropy score for each TF analyzed. TFBS entropy is measured for each TF using all the reads in each sample; thus, this analysis is not split into all exons and E1.

**Open chromatin entropy**. To assess the distribution of cfDNA fragments at open chromatin regions, consensus genomic regions from Assay for Transposase Accessible Chromatin with sequencing (ATAC-seq) data was downloaded from The Cancer Genome Atlas (TCGA) for 23 different cancer types[34]. Reads from each cfDNA sequencing sample were overlapped with these open chromatin regions to generate a set of reads for each of the 23 cancer types for each sample. For each cancer type, fragment lengths were calculated and then Shannon entropy was calculated on the frequency of fragment lengths to yield one feature for each of the 23 cancer types. Open chromatin (ATAC) entropy is measured for each cancer type in the TCGA database stated above using all the reads in each sample; thus, this analysis is not split into all exons and E1.

**In silico down-sampling in the GRAIL dataset**
The cfDNA fragment BED files were down-sampled using the Linux 'shuf' command and specifying the number of reads to select using the -n flag. Each cancer or healthy sample was independently down-sampled to 50, 25, 10, 5, and 1 million reads, while the healthy samples were independently down-sampled to 50, 75, 90, 95, and 99 million reads. These reads were then combined in every cancer: healthy and healthy: healthy combination in the five different ratios to yield a total of 100 million reads for each sample. The end result is five separate ratio samples for each cancer: healthy and healthy: healthy combination – 50:50, 25:75, 10:90, 5:95, and 1:99. Fragmentomics metrics were then calculated for each of the mixed samples. Down-sampling was also performed for the purposes of assessing the impact of overall sequencing depth to 100, 50, 25, 10, 5, and 1 million reads.

**Reporting summary**
Further information on research design is available in the Nature Portfolio Reporting Summary linked to this article.

## Data availability
Sequencing data from the GRAIL dataset is available through the European Genome-phenome Archive (ID EGAD00001005302). The institutional protocol used for the samples in this study does not allow for unrestricted access to the raw sequencing data. Requests for the raw cfDNA data must be submitted to the University of Wisconsin for review and approval. These data can be used for academic, non-commercial purposes to re-produce the results shown herein. Requests typically take a few months to review and execute. Data sharing requests for the samples from the two ongoing clinical trials (NCT03090165, NCT03725761) must be submitted to the trial organizers for review and approval. However, the fragment size/length/position data used in this study have been uploaded to FigShare (https://doi.org/10.6084/m9.figshare.28611500).

## Code availability
Code is available at https://github.com/Zhao-Lab-UW-DHO/fragmentomics_metrics/.

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

## Acknowledgements

We would like to greatly thank all the patients who participated in this study. We would also like to acknowledge Katie Andersen, the UWCCC Biospecimen collection team, and the Big Ten Cancer Research Consortium for their assistance in the work. We would like to acknowledge funding from the National Institutes of Health (DP2 OD030734 to SGZ, 1UH2CA260389 to SGZ, AJA, JML, R01CA247479 to JML, P50CA269011 to JML, SGZ, MNS, JF, 1R01CA296602-01 to SGZ and JML), Department of Defense (PC190039, PC200334, PC240611 to SGZ, JML, PC180469 to JML, PC220240 to MNS), Prostate Cancer Foundation (Movember Foundation PCF Challenge Award to JML, 2023 Debbie & Mark Attanasio Young Investigator Award to AT, 2022 Point Biopharma Young VAlor Investigator Award to MNS, 2021 Michael and Patricia Berns Young Investigator Award to MS, Tactical Award 23TACT01 to JML and FYF), the Wisconsin Alumni Research Foundation (Accelerator award to SGZ), the Doris Duke Charitable Foundation (Physician Scientist Fellowship #2021088 to MNS) and the Mary Kay Ash Foundation (Cancer Research Project #10-23 to MNS). Additional funding for this project was provided by the UW School of Medicine and Public Health from the Wisconsin Partnership Program (New Investigator Program Award #6080 to MNS). Shared research services at the UWCCC are supported by Cancer Center Support Grant (grant number P30 CA014520).

## Author contributions

Conception or design: K.T.H., M.N.S., J.M.L., S.G.Z. Acquisition, analysis, or interpretation of data: N.S., D.K., C.E.K., M.B., G.B., J.F., A.J.A., H.B., R.R.M., F.Y.F., R.O.R., K.B.W., H.E., M.S., J.M.S., M.L.B., S.R.R., A.T., K.R.K., H.K., J.S., A.W.W., M.R.C. Drafted the work or substantively revised it: KTH, MNS, JML, SGZ.

## Competing interests

The Wisconsin Alumni Research Foundation have filed a patent application (cfDNA fragmentomic detection of cancer; US20240145038A1) on which KTH and SGZ are inventors relevant to this work. KTH has a family member who is an employee of Epic Systems. SGZ has patent applications unrelated to this work with Veracyte, and a spouse who is an employee of Artera with stock, and stock in Exact Sciences. MNS reports institutional research support from Novartis unrelated to this work. MS reports speakers fees from Astellas and advisory board activity for Veracyte/Adelphi Targis. The remaining authors declare no competing interests.
