## [Peer Review File · Nature Communications]

Analysis of cfDNA Fragmentomics Metrics and Commercial Targeted Sequencing Panels

Corresponding Author: Dr Shuang Zhao

Version 0:

Reviewer comments:

Reviewer #1

(Remarks to the Author)

In this manuscript, Helzer and Sharifi et al. report a systematic comparison of fragmentomics methods utilizing existing data from targeted deep sequencing of cfDNA. Overall, this study is well executed and quite interesting. Although its methodology is not particularly novel, it performs important benchmarking and comparison work that has been lacking in the field of fragmentomics. I have several suggestions to strengthen the manuscript.

- The code that was used to produce these analyses needs to be provided, e.g. in a github repository. Although this paper uses published approaches, there are many parameter choices and possible variation in software versions that could influence the results. For transparency and reproducibility, I believe it's very important to provide the code that was used here, even if it was cloned from existing repositories with little/no modification.
- Can the authors clarify: do the datasets included in this analysis overlap with data that were originally used to benchmark their previously published entropy method? (<https://pubmed.ncbi.nlm.nih.gov/37330052/>). If so, there's a risk that hyperparameters / design choices are overfit to this dataset and may not work as well when applied to external datasets.
- The authors test performance through cross-validation. Particularly for fragmentomics features, which tend to be less interpretable than features like mutations or DNA methylation, there is still a risk of overfitting with this approach. A more convincing analysis would be to train a model on a couple of datasets with cross-validation, lock in all parameters, then test the model on unseen datasets. This is important because even with cross-validation, a model can pick up on properties between cases and controls that are systematically different but that do not generalize to other datasets.
- There are several classes of information one might want to infer from fragmentomics data. This paper focuses on distinguishing cancer types, mainly, which might be the least interesting since there are several ways to do this with high sensitivity. Ideally the authors would also look at the ability of each method to infer activity of sets of genes promoters or regulatory elements (e.g., TF binding sites). Not all methods would be expected to perform well at this task, but it would be an interesting comparison. For example, how does TF binding site entropy compare to nucleosome depletion (ie the Griffin method) for inferring activity of TFs that are expected to be active in a given cancer type?
- The authors state "the institutional protocol used for the samples in this study does not allow for unrestricted access to the raw sequencing data." In nearly all cases I've seen, IRBs allow the release of fragment locations aligned to the reference genome. This does not contain genetic / identifying info, but is critical for reproducing these results and maximizing the impact of this study. Can the authors upload such data to a public repository?
- The figure labels could be clearer about what classification task is being evaluated. For example, in Fig 2, is the AUROC showing the prediction of each class vs all others? (eg, correctly identifying healthy vs everything else, prostate vs everything else, etc.)? If so, aren't the datasets quite imbalanced? Can you show this as a confusion matrix as well to understand patterns in misclassification?
- The in silico experiment to measure performance as a function of ctDNA is nice. There's a missed opportunity to tackle an important question, though, I believe: how does performance vary as a function of depth of coverage? Could the authors perform a similar analysis with down-sampling total number of reads? I would be curious to know if increasing coverage continues to result in improved performance a lower and lower ctDNA, or if there is a plateau in performance that might be different for each feature. This would be helpful to the field.

Reviewer #2

(Remarks to the Author)
Review of Helzer, et al

In this manuscript, the authors address two important issues facing the circulating tumor DNA field: 1) relative performance of diverse published fragmentomic approaches to infer cancer site of origin (or subtype); 2) performance of fragmentomic approaches on targeted panel sequencing. The authors evaluate 12 separate metrics/methodologies on two separate cohorts (a locally sequenced cohort and publicly-available GRail cohort). They assess performance to identify various origin tissues (normal, prostate, breast incl by ER status, bladder, etc). In addition, they sampled genes in commercial panels and used in silico mixing approaches with normal free DNA to represent low tumor fractions.

The strengths of the study include addressing two highly relevant questions, strong cohorts for evaluation, incorporating diverse methodologies, and clearly presented results. The limitations of the study include difficulty tracking exactly what was tested, a missed opportunity for integration across methods which would be highly innovative, and need for more clear conclusions.

Overall, this study is highly relevant because as the ctDNA field has matured, interest in subtyping based on minimally invasive approaches has grown. Further, most current standard-of-care sequencing is targeted panel-based and if similar data to genome-based sequencing can be gleaned from panel approaches, it enhances application. The authors address diverse methodologies, two separate cohorts, and varied tumor fraction depths. To enhance its novelty, advancing this manuscript beyond a technical validation/comparison of existing methods into 1) a new integrated method; and 2) more firm conclusions and less descriptions are two potential approaches.

MAJOR COMMENTS

-Methodology naming: The methodology naming is difficult to follow through the manuscript. For example, in the first paragraph of results (line 100-125), 5 metrics are noted. In the methods, six metrics are described (line 313). Some abbreviations were used but not consistently. No description of the different methods is included in the figure legends. For clarity, it is important for the metrics to be clearly defined and named, then a constant naming convention carried through the manuscript

-Integrated method: The technical elements of the manuscript are strong, however, it seems investigating an integrated method is a missed opportunity. For example, the authors identify that certain methods perform better in certain settings (e.g. line 142 the top performing metric for SCLC in the UW cohort was MDS with an average AUROC of 0.9). Leveraging machine learning to generate an innovative cross-method approach in this strong set of data could offer a truly innovative output.

-Conclusions: The authors make few firm conclusions, despite rigorous testing. While the authors *describe* notable differences, this does not really capture true conclusions. (e.g. "even at this low ctDNA fraction, all exon depth, full gene depth, and TFBS entropy maintained AUROCs >0.74 for differentiating healthy vs. cancer" or "FoundationOne Liquid CDx panel yielded the best overall performance of the three commercial panels tested"). For example, the authors could note which algorithms differed 1) outside of confidence intervals; and 2) outside of what may be computationally relevant. Further, what types of approaches (either WGS vs. targeted, relevant genes to include in panels, or which method) are seen most important for specific settings.

MINOR COMMENTS

-While methods are clearly described, suggest that authors provide the code for reproducing the figures (even if just for the publicly-available GRail cohort) as part of the supplementary files.

Reviewer #3

(Remarks to the Author)

Reviewer #4

(Remarks to the Author)

The study gives an overview of the performance of different ctDNA fragmentomics features when applied to deep-sequenced gene panels rather than WGS data. I think it is an interesting study, but I think some important details are missing, mostly from the methods section.

1) Fragmentomics methods are much more sensitive to differences in storing, handling, or sequencing between samples than methods looking at called variants. So, it would be good to clarify if there are any such differences between the cases and healthy controls in the study.

1a) Was the sequencing order randomized, or were the cases and controls in separate batches?

1b) I can see that two different sequencing machines were used; how are the cases and controls split between the two?

1c) What is the distribution of the average number of mapped reads after deduplication in cases and controls?

1d) Did you correct for GC-bias when you calculated the depth, and are there differences in the GC-bias between cases and controls?

2) From the description, it is not precisely clear to me how you selected the alpha and lambda parameters in the glmnet. And the code is not available so I could check the details. Did you do nested cross-validation? If so, how many folds are there in the inner-fold?

3) Looking at your earlier article, I can see that you filtered based on the proper-paired flag set by the read-mapper. This could give problems for the fragment size analysis since different size filters would be applied for different samples. Have you checked what minimum and maximum fragment length cutoff this results in for each sample and how it affects your analysis?

4) For the TFBS entropy analysis, I think you could add some details. Do you use the meta clusters file from the db? Do you use all transcription factors or select a subset?

5) I am not sure about what training sets are used for which model in the results. Is "Normal" a logistic regression case/control? Is the panel with "Prostate" results based on a logistic regression model trained on prostate vs. other cases? or prostate vs controls? Or a multiclass model trained on different case labels?

Version 1:

Reviewer comments:

Reviewer #1

(Remarks to the Author)

The authors have adequately addressed my comments. I am satisfied.

(Remarks on code availability)

NA

Reviewer #2

(Remarks to the Author)

The authors have sufficiently addressed my critiques. The revisions substantially improve the clarity of the manuscript and its conclusions. While this manuscript remains primarily descriptive, the strengths remain.

(Remarks on code availability)

Superficial (e.g. not fully re-running) code appears to be complete and allow reproduction of relevant analyses/figures.

Reviewer #4

(Remarks to the Author)

My concerns about the lack of sufficient description of the methods have been addressed. Wrt. sample differences there are some, but I realize that it is hard to do get completely matched controls.

However, regarding point 3, you have not tested the minimum and maximum length of fragments in the different samples.

In the answer to the third point I raised, you write:

"Proper pairing of mapped reads is the only way to accurately measure the length of a cfDNA fragment that is longer than one read length. We are unaware of any way this information could be inferred from just a single read in a pair. This does not result in hard minimum or maximum fragment length cutoffs."

I am sorry to be pedantic, but this is incorrect. There is a difference between paired reads and "proper pair" reads. The "proper pair" flag in the bam file is defined differently for different read mappers, and a read can be part of a pair that maps to the same chromosome in the right order and still not be flagged as properly paired. In this article you use "bwa mem" and according to the bwa documentation, it performs the following calculation:

"BWA estimates the insert size distribution per 256×1024 read pairs. It first collects pairs of reads with both ends mapped with a single-end quality 20 or higher and then calculates median (Q2), lower and higher quartile (Q1 and Q3). It estimates the mean and the variance of the insert size distribution from pairs whose insert sizes are within interval $[Q1 - 2(Q3 - Q1), Q3 + 2(Q3 - Q1)]$. The maximum distance x for a pair considered to be properly paired (SAM flag 0x2) is calculated by solving equation $\Phi((x - \mu) / \sigma) = x / L * p_0$, where μ is the mean, σ is the standard error of the insert size distribution, L is the length of the genome, p_0 is prior of anomalous pair and $\Phi()$ is the standard cumulative distribution function. For mapping Illumina short-insert reads to the human genome, x is about 6-7 sigma away from the mean. Quartiles, mean, variance and x will be printed to the standard error output."

This filtering of reads based on the "proper pair" flag makes complete sense in routine sequencing, where the DNA is fragmented in the lab, and the fragment length distribution can be estimated with a normal distribution. It is, however, problematic for cfDNA where the fragment length distribution is multimodal with modes corresponding to one and two nucleosomes. If there is a large di-nucleosome peak it can lead to a higher mean and result in small fragments not being flagged as properly paired. To avoid biased results you either need to check what the min and max fragment-length of

proper-paired fragments are and the apply the most conservative filters to all the samples or you need to not filter based on the “proper pair” flag in ther bam file.

(Remarks on code availability)

Reviewer #1 (Remarks to the Author): computational expertise in DNA fragmentation

In this manuscript, Helzer and Sharifi et al. report a systematic comparison of fragmentomics methods utilizing existing data from targeted deep sequencing of cfDNA. Overall, this study is well executed and quite interesting. Although its methodology is not particularly novel, it performs important benchmarking and comparison work that has been lacking in the field of fragmentomics. I have several suggestions to strengthen the manuscript.

- The code that was used to produce these analyses needs to be provided, e.g. in a github repository. Although this paper uses published approaches, there are many parameter choices and possible variation in software versions that could influence the results. For transparency and reproducibility, I believe it's very important to provide the code that was used here, even if it was cloned from existing repositories with little/no modification.

We have uploaded the code to github, and added this link to the "Data availability" section of the methods: https://github.com/Zhao-Lab-UW-DHO/fragmentomics_metrics/

- Can the authors clarify: do the datasets included in this analysis overlap with data that were originally used to benchmark their previously published entropy method? (<https://pubmed.ncbi.nlm.nih.gov/37330052/>). If so, there's a risk that hyperparameters / design choices are overfit to this dataset and may not work as well when applied to external datasets.

The datasets in this analysis contain, and add some additional samples to, the original dataset where we exclusively looked at Shannon Entropy. We have clarified this in the "Fragmentomics Metric Performance in UW and GRAIL Cohorts" section of the results.

It is important to note that the goal of this study is not to produce and assess performance of a "locked" prediction model. Instead, we seek to compare different metrics to assess relative performance in a controlled and comprehensive manner. The absolute performance of an approach will of course depend on the exact clinical or biological question, panel design, sequencing depth, and other factors. However, relative performance can give a sense of the strengths and weaknesses of different fragmentomics metrics. We have added this context to the final paragraph of the discussion.

- The authors test performance through cross-validation. Particularly for fragmentomics features, which tend to less interpretable than features like mutations or DNA methylation, there is still a risk of overfitting with this approach. A more convincing analysis would be to train a model on a couple of datasets with cross-validation, lock in all parameters, then test the model on unseen datasets. This is important because even with cross-validation, a model can pick up on properties between cases and controls that are systematically different but that do not generalize to other datasets.

We have performed an analysis to split our cohort into a single set of training and validation sets. This was identical to the previously reported training and validation cohort used in the GRAIL cohort (70% vs. 30%). For the UW cohort, we performed a similar split. This resulted in fairly similar performance to our cross-validation results:

Response Figure 1: AUROC values for fragmentomics metric performance in the validation set after splitting the UW and GRAIL cohorts into 70% training and 30% validation each.

This is as expected, since cross-validation repeated 25 times would give a wide range of possible training and validation sets using the datasets we had. As there was no feature selection step, each cross-validation fold is essentially comprised of a training and independent validation cohort. Furthermore, each fold was identical between all metrics, so each metric is evaluated on equal footing within folds. Therefore, any potential biases should affect all the metrics, and thus our results should still help us compare the relative performance of each fragmentomics metric.

To further account for the question about generalizability, we performed the exact same steps in two different cohorts (UW and GRAIL) with two different panel designs, depths of sequencing, and a myriad of other technical differences. The fact that the results were quite similar suggests that our conclusions are consistent across datasets.

- There are several classes of information one might want to infer from fragmentomics data. This paper focuses on distinguishing cancer types, mainly, which might be the least interesting since there are several ways to do this with high sensitivity. Ideally the authors would also look at the ability of each method to infer activity of sets of genes promoters or regulatory elements (e.g., TF binding sites). Not all methods would be expected to perform well at this task, but it would be an interesting comparison. For example, how does TF binding site entropy compare to nucleosome depletion (ie the Griffin method) for inferring activity of TFs that are expected to be active in a given cancer type?

The primary goal of our study was to assess the relative performance of various fragmentomics metrics on targeted cancer gene exon panels to detect biological signals. Distinguishing between cancer types is a useful metric, as there are large biological differences between cancer types and non-cancer patients that we and others have shown to be detectable using cfDNA fragmentomics.

We agree that inference of gene promoter activity or other regulatory elements such as TF binding sites is an important question. Unfortunately, these panels are focused on cancer gene coding exons and do not include promoters. This type of coding exon-based panel design is standard on commercial or clinically used targeted

cancer gene panels. These clinical panels are thus the focus of this work, as it represents what is currently being used widely in clinical practice and trials.

Thus, we cannot assess promoters, and can only assess limited TFBS's, using this type of panel. For example, in our UW panel design, only 1 entire AR, 5 entire ESR1, and 3 entire PGR TFBS's fall completely within the panel. We utilize partial overlaps for our classifiers, but while these can provide partial information about differences between samples, as the amount of overlap is consistent across samples, they may not be able to accurately infer the status of TF binding. This likely explains why the performance of TFBS entropy lags behind other metrics. The examination of promoters and other regulatory elements is an intriguing question that has been examined in depth by other excellent papers which include these regions through targeted panels or WGS (Hiatt et al. Science Advances 2024; Herberts et al. Nature 2022). We are in the process of designing a new targeted panel that also adds promoter regions, and this will be an important area of future investigation.

Per the reviewers suggestion, we ran Griffin in the UW cohort using the same TFBS's as our TFBS analysis, and examined its performance compared to the other metrics. We attempted to run Griffin on the GRAIL cohort, but ran into technical issues due to the alignment of the data to the hg19 human genome build rather than hg38. Per the Griffin documentation "Griffin has not been tested on genome builds other than hg38, but this snakemake is provided in case you would like to try a different genome build" (<https://github.com/adoebley/Griffin?tab=readme-ov-file>).

We attempted to use this script to calculate the frequency of fragments with each GC content across the mappable regions of hg19, but this required a Hoffman mappability track. The hg38 mappability track is available through the UCSC Genome Browser, but an equivalent file for the hg19 genome is missing from this source. We were able to find a mappability file from the Hoffman Lab's website (<https://bismap.hoffmanlab.org/>) for hg19, but upon download and testing of the file with the Griffin 'griffin_genome_GC_frequncy' script, we found that the file was truncated and produced an error when running. Upon investigation of the file, it was found that chr2 had been truncated mid-line (see excerpt below).

chr2	211624928	211624929	0.23
chr2	211624929	211624930	0.23
chr2	211624930	211624chr3	60012 60013 0.0
chr3	60013	60014	0.01
chr3	60014	60015	0.02
chr3	60015	60016	0.03

The final chr2 coordinate listed in this file (211624930) is approximately 31 Mbp from the end of the chromosome, so simply disregarding this line was not an option (Other chrs analyzed did not display this level of missing distance from the end: for example, chr1, chr3, and chr4, ended ~10kbp, ~62kbp, and ~110kbp from the end, many orders of magnitude less than chr2). Additionally, access to the raw fastq files is not provided for the GRAIL cohort through the EGA archive, only the hg19-aligned BAMs are available. We also considered using a liftover tool to convert the mappability track or the GRAIL BAM files to the correct genome build, but these tools tend to introduce errors which would produce inaccurate data. For these reasons, we were not able to run Griffin on the GRAIL cohort, thus limiting our analysis to the UW cohort. We have added as supplemental Figure S3 the new figure below incorporating Griffin where we find that it performs well, but not as well as our highest performing metrics.

UW

Figure S3. Overview of Fragmentomic Metric Performance in the UW Cohort including Griffin. Griffin was performed on the UW cohort as described in the report (see methods), using TFBSs from 808 TFs as the regions analyzed, and central coverage, mean coverage, and amplitude for each TF as the features for training.

- The authors state “the institutional protocol used for the samples in this study does not allow for unrestricted access to the raw sequencing data.” In nearly all cases I’ve seen, IRBs allow the release of fragment locations aligned to the reference genome. This does not contain genetic / identifying info, but is critical for reproducing these results and maximizing the impact of this study. Can the authors upload such data to a public repository?

This is a great suggestion, we have uploaded the fragment locations to FigShare to be available with the publication of this manuscript. The private link is:
<https://figshare.com/s/182a1e3c90bf979ae072>

The public link will be at: <https://doi.org/10.6084/m9.figshare.28611500>

- The figure labels could be clearer about what classification task is being evaluated. For example, in Fig 2, is the AUROC showing the prediction of each class vs all others? (eg, correctly identifying healthy vs everything else, prostate vs everything else, etc.)? If so, aren’t the datasets quite imbalanced? Can you show this as a confusion matrix as well to understand patterns in misclassification?

The reviewer is correct that the AUROCs are showing the prediction of each class vs. all others. In the training, we weight the samples inversely to the number in each class to balance the model. We have added supplemental figures showing the confusion matrix for each metric as supplemental Figure S5 and S6.

Figure S5: Confusion matrices for fragmentomics metric performance in the UW cohort. Shading represents the proportion of true (Actual) samples predicted by each model.

Figure S6: Confusion matrices for fragmentomics metric performance in the GRAIL cohort. Shading represents the proportion of true (Actual) samples predicted by each model.

- The *in silico* experiment to measure performance as a function of ctDNA is nice. There's a missed opportunity to tackle an important question, though, I believe: how does performance vary as a function of depth of coverage? Could the authors perform a similar analysis with down-sampling total number of reads? I would be curious to know if increasing coverage continues to result in improved performance a lower and lower ctDNA, or if there is a plateau in performance that might be different for each feature. This would be helpful to the field.

This is an excellent question. We have performed down-sampling of the total number of reads and evaluated performance as a function of depth for 100M, 50M, 25M, 10M, 5M, and 1M reads. For this analysis, we focused on the GRAIL cohort given the higher depth of sequencing. As expected, there is some decrease in performance as the read numbers become very small, though the magnitude of the effect varies depending on the metric. These results have been added as supplemental Figure S7.

Figure S7: Down-sampling fragmentomics metric performance in the GRAIL cohort. Each sample in the GRAIL cohort was down-sampled to the level indicated (in millions of reads), and the fragmentomics metrics were calculated as previously described. These metrics were tested for their ability to predict three different cancer types along with healthy vs. cancer using a GLMnet machine learning model with 10-fold cross validation. Five independent replicates of down-sampling and twenty-five replicates of 10-fold cross validation within each down-sampling replicate were performed and boxplots of the AUROC for each metric are shown.

Reviewer #2 (Remarks to the Author): clinical expertise in cfDNA analysis

Review of Helzer, et al

In this manuscript, the authors address two important issues facing the circulating tumor DNA field: 1) relative performance of diverse published fragmentomic approaches to infer cancer site of origin (or subtype); 2) performance of fragmentomic approaches on targeted panel sequencing. The authors evaluate 12 separate metrics/methodologies on two separate cohorts (a locally sequenced cohort and publicly-available GRAIL cohort). They assess performance to identify various origin tissues (normal, prostate, breast incl by ER status, bladder, etc). In addition, they sampled genes in commercial panels and used in silico mixing approaches with normal free DNA to represent low tumor fractions.

The strengths of the study include addressing two highly relevant questions, strong cohorts for evaluation, incorporating diverse methodologies, and clearly presented results. The limitations of the study include difficulty tracking exactly what was tested, a missed opportunity for integration across methods which would be highly innovative, and need for more clear conclusions.

Overall, this study is highly relevant because as the ctDNA field has matured, interest in subtyping based on minimally invasive approaches has grown. Further, most current standard-of-care sequencing is targeted panel-based and if similar data to genome-based sequencing can be gleaned from panel approaches, it enhances application. They authors address diverse methodologies, two separate cohorts, and varied tumor fraction depths. To enhance its novelty, advancing this manuscript beyond a technical validation/comparison of existing methods into 1) a new integrated method; and 2) more firm conclusions and less descriptions are two potential approaches.

MAJOR COMMENTS

-Methodology naming: The methodology naming is difficult to follow through the manuscript. For example, in the first paragraph of results (line 100-125), 5 metrics are noted. In the methods, six metrics are described (line 313). Some abbreviations were used but not consistently. No description of the different methods is included in the figure legends. For clarity, it is important for the metrics to be clearly defined and named, then a constant naming convention carried through the manuscript

We apologize for the confusion. This first paragraph of the results describes the five broad classes of metrics, within which multiple specific metrics exist. A total of 13 fragmentomics metrics were assessed: normalized depth at (1) all exons individually, (2) full gene, (3) E1, Shannon entropy at (4) all exons, (5) E1, MDS at (6) all exons, (7) E1, small fragments at (8) all exons, (9) E1, (10) fragment bins, (11) TFBS entropy, (12) ATAC entropy, as well as (13) all metrics combined. We have clarified this in the "Overview of Fragmentomic Analyses" section of the results and in the "Metric calculations" section of the methods.

-Integrated method: The technical elements of the manuscript are strong, however, it seems investigating an integrated method is a missed opportunity. For example, the authors identify that certain methods perform better in certain settings (e.g. line 142 the top performing metric for SCLC in the UW cohort was MDS with an average AUROC of 0.9). Leveraging machine learning to generate an innovative cross-method approach in this strong set of data could offer a truly innovative output.

We have added an analysis where we integrated all the fragmentomics metrics into a single model. Combining all metrics did not result in improved performance beyond the best individual fragmentomics metrics on the full panels. However, there was a very small increase in performance on the commercial panel gene subsets, suggesting that perhaps merging all metrics can provide some additional information in smaller panels. We have added this analysis to Figures 2-5.

UW

Updated Figure 2 incorporating the combined metric (All Combined)

GRAIL

Updated Figure 3 incorporating the combined metric (All Combined)

Updated Figure 4A incorporating the combined metric (All Combined)

UW Cohort

Updated Figure 4B incorporating the combined metric (All Combined)

Updated Figure 5A incorporating the combined metric (All Combined)

GRAIL Cohort

Updated Figure 5B incorporating the combined metric (All Combined)

*-Conclusions: The authors make few firm conclusions, despite rigorous testing. While the authors *describe* notable differences, this does not really capture true conclusions. (e.g. “even at this low ctDNA fraction, all exon depth, full gene depth, and TFBS entropy maintained AUROCs >0.74 for differentiating healthy vs. cancer” or “FoundationOne Liquid CDx panel yielded the best overall performance of the three commercial panels tested”). For example, the authors could note which algorithms differed 1) outside of confidence intervals; and 2) outside of what may be computationally relevant. Further, what types of approaches (either WGS vs. targeted, relevant genes to include in panels, or which method) are seem most important for specific settings.*

We have added several additional conclusions to the discussion section of our manuscript per the reviewer suggestions:

- ATAC Entropy consistently underperforms the other metrics in both panels and all classes. This is likely due to less information being contained within ATAC peaks compared to all exons, especially since exons are not particularly enriched for regulatory regions.
- In general, depth-based metrics and utilizing all exons performed better, though there were certain cancer types such as SCLC where MDS of all exons seemed to perform remarkably well.
- Most current targeted panels are based around cancer gene coding exons as they are primarily looking for DNA variants. Our work suggests that adding genes whose regulation/expression is of interest would allow for additional fragmentomic analysis.
- Combining all metrics into a single integrated model did not improve performance over the best individual metrics in the full panels, though there was perhaps a small improvement in performance in the smaller commercial panel gene subsets.

MINOR COMMENTS

-While methods are clearly described, suggest that authors provide the code for reproducing the figures (even if just for the publicly-available GRAIL cohort) as part of the supplementary files.

We have uploaded the code to github, and added this link to the data availability section of the manuscript: https://github.com/Zhao-Lab-UW-DHO/fragmentomics_metrics/

Reviewer #3 (Remarks to the Author): ECR co-review

Reviewer #4 (Remarks to the Author): expertise in cfDNA fragmentomics wet lab methods

The study gives an overview of the performance of different ctDNA fragmentomics features when applied to deep-sequenced gene panels rather than WGS data. I think it is an interesting study, but I think some important details are missing, mostly from the methods section.

1) Fragmentomics methods are much more sensitive to differences in storing, handling, or sequencing between samples than methods looking at called variants. So, it would be good to clarify if there are any such differences between the cases and healthy controls in the study.

1a) Was the sequencing order randomized, or were the cases and controls in separate batches?

As samples were processed over several years, sequencing was done whenever a sufficient number of samples were available. 97% of samples were processed in batches that contained multiple phenotypes, though not all phenotypes were represented in every batch. All samples were processed in an identical manner using the exact same protocol. We have clarified this in the “Sequencing and processing of UW samples” section of the methods.

1b) I can see that two different sequencing machines were used; how are the cases and controls split between the two?

All sequencing for the UW cohort was performed on a Novaseq 6000 at our institutional sequencing core. The GRAIL cohort was downloaded from EGAS00001003755. Per their publication, this was sequenced on an Illumina HiSeq X. However, these datasets were analyzed independently and not directly compared as there are other differences such as panel design. We have clarified these details in the “Sequencing and processing of UW samples” section of the methods.

1c) What is the distribution of the average number of mapped reads after deduplication in cases and controls?

We have generated a boxplot showing the distribution of the average number of mapped reads after deduplication across each phenotype class, and added this as supplemental Figure S2. The depth-based metrics we used are normalized for the total sequencing depth of each sample.

Figure S2. Total sequencing reads post-deduplication in each cohort by phenotype. NEPC, neuroendocrine prostate cancer; ERpos, ER-positive breast cancer; ERneg, ER-negative breast cancer; NSCLC, non-small cell lung carcinoma; SCLC, small cell lung carcinoma; RCC, renal cell carcinoma.

1d) Did you correct for GC-bias when you calculated the depth, and are there differences in the GC-bias between cases and controls?

GC-bias is an important factor for WGS fragmentomics. However, as we were looking at primarily exonic regions of important protein coding genes, the GC content does not fluctuate as much as across the whole genome. We investigated this further, and found that that the GC content of each exon is not very strongly correlated with the various fragmentomics metrics. We have added this analysis as supplemental Figure S2. Furthermore, as we are comparing across samples, rather than comparing across regions in the genome, any minor GC-bias should affect the same region in each sample equally.

Figure S2. Correlation of GC content and fragmentomic metrics. For each genomic region tested in each feature, the GC content was calculated and plotted against the feature metric across all samples. A linear model was fit to the data and the square of the Pearson correlation coefficient is reported (R^2).

2) From the description, it is not precisely clear to me how you selected the alpha and lambda parameters in the glmnet. And the code is not available so I could check the details. Did you do nested cross-validation? If so, how many folds are there in the inner-fold?

We used latin-hypercube with 10 parameters, and performed nested cross-validation with 5 folds in the inner-fold and used the same approach for all fragmentomics metrics so that the results would be comparable. We have clarified this in the “Machine Learning” section of the methods.

We have uploaded the code to github, and added this link to the “Data availability” section of the methods: https://github.com/Zhao-Lab-UW-DHO/fragmentomics_metrics/

3) Looking at your earlier article, I can see that you filtered based on the proper-paired flag set by the read-mapper. This could give problems for the fragment size analysis since different size filters would be applied for

different samples. Have you checked what minimum and maximum fragment length cutoff this results in for each sample and how it affects your analysis?

Proper pairing of mapped reads is the only way to accurately measure the length of a cfDNA fragment that is longer than one read length. We are unaware of any way this information could be inferred from just a single read in a pair. This does not result in hard minimum or maximum fragment length cutoffs.

4) For the TFBS entropy analysis, I think you could add some details. Do you use the meta clusters file from the db? Do you use all transcription factors or select a subset?

We apologies for the lack of clarity for the TFBS entropy analysis section. We have clarified this in the methods:

“To assess the distribution of cfDNA fragments at transcript factor binding sites (TFBS), a collection of consensus *Homo sapiens* TFBSs was downloaded from the Gene Transcription Regulation Database³⁶ (GTRD, v19.10). Specifically, the ‘Homo_sapiens_meta_clusters.interval’ file was downloaded. For each TF, the top 5000 sites with the greatest amount of experimental support (exp.count column) were used for analysis, which is defined as the number of experiments which detected the binding site. TFs with fewer than 5000 sites were discarded, leaving a total of 808 TFs used for the analysis. Read fragments from each cfDNA sample were then overlapped with the curated TFBSs, and Shannon entropy was calculated on the frequencies of the fragment sizes overlapping sets of TFBSs. This yielded an entropy score for each TF analyzed.”

5) I am not sure about what training sets are used for which model in the results. Is “Normal” a logistic regression case/control? Is the panel with “Prostate” results based on a logistic regression model trained on prostate vs. other cases? or prostate vs controls? Or a multiclass model trained on different case labels?

We apologize for the confusion. 10-fold cross-validation was performed using 25 unique seeds to result in 25 unique sets of 10 folds. Folds were set first, so that the same folds and therefore training/validation were used for all fragmentomics metrics making direct comparison possible. All models reported in the manuscript are trained identically as a single multi-class logistic regression model in order to make the results comparable across metrics. We have clarified this in the “Machine Learning” section of the methods.

REVIEWERS' COMMENTS

Reviewer #1 (Remarks to the Author):

The authors have adequately addressed my comments. I am satisfied.

Reviewer #1 (Remarks on code availability):

NA

Reviewer #2 (Remarks to the Author):

The authors have sufficiently addressed my critiques. The revisions substantially improve the clarity of the manuscript and its conclusions. While this manuscript remains primarily descriptive, the strengths remain.

Reviewer #2 (Remarks on code availability):

Superficial (e.g. not fully re-running) code appears to be complete and allow reproduction of relevant analyses/figures.

Reviewer #4 (Remarks to the Author):

My concerns about the lack of sufficient description of the methods have been addressed. Wrt. sample differences there are some, but I realize that it is hard to do get completely matched controls. However, regarding point 3, you have not tested the minimum and maximum length of fragments in the different samples.

In the answer to the third point I raised, you write:

“Proper pairing of mapped reads is the only way to accurately measure the length of a cfDNA fragment that is longer than one read length. We are unaware of any way this information could be inferred from just a single read in a pair. This does not result in hard minimum or maximum fragment length cutoffs.”

I am sorry to be pedantic, but this is incorrect. There is a difference between paired reads and “proper pair” reads. The “proper pair” flag in the bam file is defined differently for different read mappers, and a read can be part of a pair that maps to the same chromosome in the right order and still not be flagged as properly paired. In this article you use “bwa mem” and according to the bwa documentation, it performs the following calculation:

*“BWA estimates the insert size distribution per 256*1024 read pairs. It first collects pairs of reads with both ends mapped with a single-end quality 20 or higher and then calculates median (Q2), lower and higher quartile (Q1 and Q3). It estimates the mean and the variance of the insert size distribution from pairs whose insert sizes are within interval $[Q1-2(Q3-Q1), Q3+2(Q3-Q1)]$. The maximum distance x for a pair considered to be properly paired (SAM flag 0x2) is calculated by solving equation $\Phi((x-\mu)/\sigma)=x/L * p_0$, where μ is the mean, σ is the standard error of the insert size distribution, L is the length of the genome, p_0 is prior of anomalous pair and $\Phi()$ is the standard cumulative distribution function. For mapping Illumina short-insert reads to the human genome, x is about 6-7 σ away from the mean. Quartiles, mean, variance and x will be printed to the standard error output.”*

This filtering of reads based on the “proper pair” flag makes complete sense in routine sequencing, where the DNA is fragmented in the lab, and the fragment length distribution can be estimated with a normal distribution. It is, however, problematic for cfDNA where the fragment length distribution is multimodal with modes corresponding to one and two nucleosomes. If there is a large di-nucleosome peak it can lead to a higher mean and result in small fragments not being flagged as properly paired. To avoid biased results you either

need to check what the min and max fragment-length of proper-paired fragments are and the apply the most conservative filters to all the samples or you need to not filter based on the “proper pair” flag in ther bam file.

We thank the reviewer for providing more detail on their question about proper paired reads. The reviewer is correct, that proper paired reads were filtered out by default by the original de-duplication tool used (Conner). Thus, we switched to a different de-duplication tool (UMI tools) and retained all paired reads regardless of the “proper pair” flag. As the reviewer suggested, we then also applied a conservative uniform standard size filter removing reads <20bp or >500bp. As this slightly modified the read fragments for every sample, we then re-calculated all fragmentomic metrics, and re-performed all machine learning, mixing, and down-sampling analyses. This resulted in minor changes in the AUC values throughout the manuscript, but did not change any of the conclusions of the manuscript. We have updated all figures/tables/supplemental data. We have also uploaded the updated fragment size/location files to FigShare.